

# Gender-specific associations between apolipoprotein A1 and arterial stiffness in patients with nonalcoholic fatty liver disease

Xulong Sun[1,2,*], Ruifang Chen[1,*], Guangyu Yan[1], Zhiheng Chen[3], Hong Yuan[1], Wei Huang[1] and Yao Lu[1,4]

[1] Clinical Research Center, The Third Xiangya Hospital of Central South University, Changsha, China
[2] Department of General Surgery, The Third Xiangya Hospital of Central South University, Changsha, China
[3] Health Management Center, The Third Xiangya Hospital of Central South University, Changsha, China
[4] Department of Life Science and Medicine, King's College London, London, UK
* These authors contributed equally to this work.

Corresponding authors
Wei Huang, 118312057@csu.edu.cn
Yao Lu, yao.lu@kcl.ac.uk

## ABSTRACT

**Background:** Lipid metabolism factors may play an important role in the progression of nonalcoholic fatty liver disease (NAFLD) and its related cardiovascular dysfunctions. The study aims to assess whether Apolipoprotein A-1 (ApoA1) was associated with vascular stiffness in NAFLD patients.

**Methods:** From 2012 to 2013, we included 2,295 non-alcohol users with fatty liver disease (1,306 male patients) and completely excluded subjects who drank any alcohol ever to eliminate the effect of alcohol intake. The serum ApoA1 levels and the brachial-ankle pulse wave velocity (baPWV) were measured.

**Results:** The baPWV in men was much higher than in female patients (1,412.79 cm/s vs. 1,358.69 cm/s, $P < 0.001$). ApoA1 level was positively associated with baPWV odd ratio (OR), 4.18; 95% confidence interval (CI) [1.16–15.1], $P < 0.05$) in patients with AST/ALT < 1 and (OR, 4.70; 95% CI [1.36–16.23], $P < 0.05$) in patients with AST/ALT ≥ 1 respectively. Only arterial stiffness in men was associated with ApoA1 (OR, 3.96; 95% CI [1.29–12.30], $P < 0.05$) in logistics regression models adjusted for age, gender, body mass index, education attainment, physical activity, smoking, history of hypertension and high-density lipoprotein. The relationship between ApoA1 and baPWV in male NAFLD patients remained significant (confidence, 156.42; 95% CI [49.34–263.50], $P < 0.05$) in the fully adjusted linear regression model.

**Conclusion:** The serum ApoA1 was associated with arterial stiffness in male NAFLD patients. Increased ApoA1 level should be considered as an independent risk factor for arterial stiffness in male NAFLD patients, suggesting that NAFLD may alter arterial stiffness by "ApoA1-related" mechanism in men.

## INTRODUCTION

Nonalcoholic fatty liver disease (NAFLD) is increasingly recognized as a major chronic liver disease not only in Western countries but also in Asian countries like China in which 15–30% of the general adult population suffers from NAFLD (*Wong, 2013*). NAFLD, which is characterized by the accumulation of triglycerides in hepatocytes in the absence of excessive alcohol intake and considered as systemic consequence (*Byrne & Targher, 2015*; *Francque, Der Graaff & Kwanten, 2016*), has been confirmed to be associated with an increased risk of cardiovascular disease (CVD) and the incidence of cardiovascular events (*De Alwis & Day, 2008*). There is substantial evidence that NAFLD is associated with increased coronary artery calcification, increased pulse wave velocity (PWV), increased coronary membrane thickness, endothelial dysfunction, all of which are established markers of CVD (*Fan et al., 2016*; *Iannaccone et al., 2007*; *Vlachopoulos et al., 2010*). More specifically, evidence NAFLD begins with the aberrant accumulation of triglyceride in the liver, thus lipid metabolism disorders play an important role in the disease progression as well as its complications like vascular dysfunction and related CVD outcomes (*Francque, Der Graaff & Kwanten, 2016*). Therefore, it is vital to identify the potential risk or biomarkers for the prevalence of cardiovascular diseases and related complications like arterial stiffness and vascular dysfunction, especially in NAFLD patients.

Pulse wave velocity is a valuable tool for assessing arterial stiffness by measuring the pulse propagation time and the travel distance between the studied arterial site, which is associated to abnormal vascular tone, thickening of smooth muscle and changes in blood viscosity (*Deloach & Townsend, 2008*; *Kelly et al., 1989*). Therefore, PWV is considered as a important marker of CVD events (*Bae et al., 2013*). Nevertheless, aortic stiffness modulation remains a strong target for therapeutic intervention, as it is the most powerful cardiovascular risk factor indeed a surrogate outcome rather than a risk factor.

Arterial stiffness could be progressive due to lipid metabolism (*Fok & Cruickshank, 2015*). Lipoprotein metabolism is usually thought to be associated with arterial changes, which will be of great interest in knowing how and to what extent arterial stiffness conditions are associated with CVD events in NAFLD patients (*Sorokin & Kotani, 2015*). Patients with NAFLD often have imbalances in lipid metabolism resulting in elevated very low lipoprotein and decreased high-density lipoprotein (HDL) levels as well as low-density lipoprotein (LDL). Therefore, more lipid biomarkers should be identified to predict vascular dysfunctions. Recent studies forced on the effect HDL and LDL on arterial stiffness (*Caldas et al., 2013*; *Gordon et al., 2013*), but few have concentrated on the protein components of these lipoproteins, especially in NAFLD patients. Apolipoprotein A-1 (ApoA1) is the major protein component of HDL, which plays an essential role in lipid production and in an important modulator of processes associated with glucose intolerance and diet-induced NAFLD development (*Castellani & Lusis, 2001*; *Smith, 2010*). Lower ApoA1 has been reported to be statistically significant associations with high-risk atherosclerosis (*Zhao et al., 2014*). Nevertheless, the relationship between ApoA1 and PWV remains unknown.

In this study, we addressed the question of whether the serum ApoA1 concentration was associated with or was a risk factor for NAFLD and whether ApoA1 was associated with arterial stiffness; also, the relationship between ApoA1 and PWV in NAFLD was examined.

## MATERIALS AND METHODS

People were recruited in the Health Management Center of The Third Xiangya Hospital from March 2012 to December 2013. The study was approved by the Institutional Review Board of Central South University (R18030) and informed oral consent was obtained from all subjects. Evidence including non-alcoholic users NAFLD and NAFLD is based on the following items: (1) The diffusion of near-field echoination in the hepatic region increases (stronger than that in the kidney and spleen) and the far field echo is gradual attenuated; (2) Intra-hepatic lacuna structure is not clear; (3) Mild to moderate hepatomegaly, with round and blunt border; (4) Color Doppler ultrasound shows a decrease or even difficulty in showing blood flow signals in the liver, but the distribution of blood flow is normal. If any one or more matches in items 1 and 2–4 are made, the NAFLD is diagnosed. Hepatic ultrasound is carried out blindly by trained ultrasoundists (*De Zeng et al., 2008*).

The exclusion criteria were as follows: (1) patients with hepatotoxic drug treatment; (2) history of heart failure, stroke, arteriosclerosis, and diabetes; (3) participants with hepatitis C virus (HCV), hepatitis B virus (HBV) and any chronic liver disease; (4) participants with other chronic diseases: human immunodeficiency virus (HIV), cancer and end-stage kidney disease; (5) incomplete data information.

### Clinical and laboratory assessments

Anthropometric factors include height, weight, systolic blood pressure (SBP), diastolic blood pressure (DBP) and the brachial-ankle pulse wave velocity (baPWV). Body mass index (BMI) was calculated as weight (kg) divided by height squared (m$^2$). Weight and height were measured after participants had removed their shoes and heavy clothing. They were measured in meters using a scale from the G-TECH Company. Blood pressure measurements were done with a mercury sphygmomanometer with the participants in a chair using the right arm after 5 min in rest. The average value of three readings was accepted as SBP and DBP.

Serum HDL, LDL and ApoA1 were prepared from fasting blood samples, and the supernatant was collected and stored at −20 °C and −80 °C. The serum levels of HDL, LDL were determined by enzymatic methods and the Friedwald equation and ApoA1 was measured by an immunological turbidity testing. They were measured using a biochemical detector from the HITACHI Company (7600-020).

The definition of social behavior factors has also been reported in our previous study (*Lu et al., 2015*). Educational attainment is categorized as "<college graduation", "college graduation" or ">college graduation". Physical activity and smoking had different types in the standard questionnaires. Physical activity was categorized as "Never", "Occasionally" or "Frequently (vigorous exercise at least three times per week)." Smoking

was categorized as "Quit", "Never", "Occasionally" or "Frequently". The diagnostic criteria of hypertension history were systolic or diastolic blood pressure ≥140 or ≥90 mmHg.

## baPWV measurement

The brachial-ankle pulse wave velocity values were assessed using an automatic, non-invasive vascular screening device Omron Health Medical (China) LTD., Wuhan, China calculated as the distance travelled by the pulse wave divided by the time taken to travel the distance. The baPWV values were measured by the average of the left side baPWV and right side baPWV. The baPWV value is measured by the average of the baPWV on the left and the baPWV on the right. This is obtained by measuring blood pressure levels and pulsewaves in the arms and legs with four pneumatic pressure cuffs around the four limbs. Measurements were taken between 7:30 and 9:00 in the morning on the examination day. All the data was collected after the participants had 5 min of bed rest.

Since the reference value of baPWV up to 1,400 cm/s was used as an indicator of the severity of arteriosclerosis vascular damage, increased arterial stiffness was defined as baPWV of 1,400 cm/s or higher (Yamashina et al., 2003).

## Statistical analysis

Values for continuous variables are presented as mean ± SD. Comparisons between female NAFLD patients and male NAFLD patients were performed using Pearson's Chi-square analysis for categorical variables and T-test for continuous variables. Multivariate logistic regression was used to assess the association between ApoA1 and baPWV totally and separately by gender. Linear regression analysis was used to determine the correlation between ApoA1 and baPWV in male NAFLD patients. The analysis was conducted after adjusted for age, gender, BMI, education, physical activity, smoking, history of hypertension, HDL and LDL level. As serum AST/ALT ratio is a good index to assess fibrosis advances. Patients can be divided into two groups by comparison of the ratio to 1. Adjusted odds ratio and 95% confidence intervals (CIs) were calculated. An odds ratio (OR) >1 was considered a risk factor, and an OR <1 was considered as a protective factor.

For exploratory purposes, the adjusted relationship between ApoA1 and the incidence of arterial stiffness was also assessed by a multivariate logistic regression model with restricted cubic splines. The relationships were depicted based on this model in which ApoA1 levels were categorized into 0.5 g/L increment from <1 to >3 g/L and were adjusted for age, gender, BMI, education, physical activity, smoking, history of hypertension, HDL and LDL level. Values of $P < 0.05$ were considered statistically significant. All the data analyses and figure were performed using STATA software version 14.0.

## RESULTS

A total of 2,295 subjects (1,306 males) were enrolled the baseline characteristics of the study participants grouped by gender are shown in Table 1. The participants' mean age was 48.47 ± 10.22 years old, and the average BMI was 24.8 ± 3.3 kg/m². The means of systolic and diastolic blood pressure were 124.65 ± 16.54 and 78.13 ± 11.35 mmHg.

| Table 1 Characteristics of the study participants, by gender. | | | | |
|---|---|---|---|---|
| Variable | Total (mean ± SD) N = 2,295 | Male (mean ± SD) N = 1,306 | Female (mean ± SD) N = 989 | P value |
| Age (years) | 48.47 ± 10.22 | 47.76 ± 9.92 | 49.4 ± 9.33 | 0.0001 |
| BMI | 24.80 ± 3.3 | 25.57 ± 3.14 | 23.79 ± 3.24 | <0.001 |
| SBP (mmHg) | 124.65 ± 16.54 | 126.54 ± 15.09 | 122.13 ± 17.98 | <0.001 |
| DBP (mmHg) | 78.13 ± 11.35 | 80.88 ± 11.24 | 74.5 ± 10.45 | <0.001 |
| Educational attainment (%) | | | | |
| <College graduation | 365 (15.9%) | 275 (21.1%) | 90 (9.1%) | |
| College graduation | 1481 (64.5%) | 955 (73.1%) | 526 (53.2%) | <0.001 |
| >College graduation | 449 (19.6%) | 76 (5.8%) | 373 (37.7%) | |
| Physical Activity (%) | | | | |
| Never | 251 (10.9%) | 161 (12.3%) | 90 (9.1%) | |
| Occasionally | 560 (24.4%) | 304 (23.3%) | 256 (25.9%) | |
| Frequently | 1,484 (64.7%) | 841 (64.4%) | 643 (65.0%) | 0.04 |
| Smoking Status (%) | | | | |
| Quit | 101 (4.4%) | 93 (7.1%) | 8 (0.8%) | |
| Never | 1423 (62.0%) | 548 (42.0%) | 875 (88.5%) | |
| Occasionally | 207 (9.0%) | 170 (13.0%) | 37 (3.7%) | <0.001 |
| Frequently | 564 (24.6%) | 495 (37.9%) | 69 (7.0%) | |
| History of Hypertension (%) | 246 (10.7%) | 150 (11.5%) | 96 (9.7%) | <0.001 |
| HDL (mmol/L) | 1.58 ± 0.4 | 1.43 ± 0.35 | 1.77 ± 0.39 | <0.001 |
| LDL (mmol/L) | 3.00 ± 0.88 | 3.01 ± 0.88 | 2.98 ± 0.89 | 0.54 |
| ApoA1 (g/L) | 1.51 ± 0.28 | 1.43 ± 0.26 | 1.62 ± 0.27 | <0.001 |
| Right baPWV (cm/s) | 1,388.01 ± 278.2 | 1,412.51 ± 261.31 | 1,355.66 ± 296.09 | |
| Left baPWV (cm/s) | 1,390.95 ± 277.03 | 1,413.07 ± 257.73 | 1,361.72 ± 298.24 | |
| baPWV (cm/s) | 1,389.48 ± 273.5 | 1,412.79 ± 255.13 | 1,358.69 ± 293.34 | <0.001 |
| Arterial stiffness | 953 (41.56%) | 585 (44.79%) | 350 (35.39%) | <0.001 |

Note:
Values for categorical variables are presented as number (percentage); Values for continuous variables are presented as mean ± SD.

The average HDL and LDL were 1.58 ± 0.4 mmol/L and 3 ± 0.88 mmol/L. The means of ApoA1 and baPWV were 1.51 ± 0.28 g/L and 1,389.48 ± 273.5 cm/s. There were significant differences in education attainment, smoking, and history of hypertension prevalence between male and female NAFLD patients ($P < 0.05$). Female NAFLD patients are older and had lower BMI, blood pressure, baPWV and predominantly had higher HDL and ApoA1 level ($P < 0.05$, Table 1).

The total prevalence of arterial stiffness is 41.56%, the prevalence for men is much higher than that for women (44.79% vs 35.39%, $P < 0.001$) (Table 1). A multi-logistic regression analysis was performed to reveal the relationship between ApoA1 and baPWV among NAFLD patients with different genders. The results of three different models were displayed in Table 2. In model 1, after adjusted for age, gender and BMI, only ApoA1 level in male NAFLD patients was significantly associated with baPWV (OR, 1.93; 95% CI [1.22–3.05], $P < 0.001$). In model 2 which adjusted for age, gender, BMI, education

**Table 2 Odd ratios for the incidence of arterial stiffness, by gender.**

| Variable | Total | | | | | | | Male | | Female | |
| --- | --- | --- | --- | --- | --- | --- | --- | --- | --- | --- | --- |
| | N = 2,295 | | AST/ALT< 1 | | AST/ALT≥1 | | | N = 1,306 | | N = 989 | |
| | OR | 95% CI | OR | 95% CI | OR | 95% CI | | OR | 95% CI | OR | 95% CI |
| Model 1 | 1.20 | [0.84–1.72] | 1.88* | [1.06–3.33] | 1.36 | [0.83–2.22] | | 1.93** | [1.22–3.05] | 0.57 | [0.31–1.03] |
| Model 2 | 1.69* | [1.08–2.66] | 1.89 | [0.95–3.77] | 1.80 | [0.97–3.37] | | 2.80** | [1.55–5.04] | 0.74 | [0.35–1.58] |
| Model 3 | 4.46** | [1.85–10.77] | 4.18* | [1.16–15.1] | 4.70* | [1.36–16.23] | | 3.96* | [1.29–12.30] | 2.92 | [0.63–13.59] |

Notes:
* $p < 0.05$.
** $p < 0.001$.
Model 1: logistic regression for the Incidence of arterial stiffness and ApoA1, and adjusted for age, gender and BMI for total population; age and BMI for Male and Female population.
Model 2: Model 1 and adjusted for educational attainment, physical activity, smoking status and history of hypertension.
Model 3: Model 2 and adjusted for HDL LDL.

**Table 3 Linear regression between Apoa1 and baPWV in male patients.**

| Variable | Coefficient | 95% CI | P value |
| --- | --- | --- | --- |
| Model 1 | 160.02 | [107.37–212.66] | <0.001 |
| Model 2 | 139.84 | [83.97–195.68] | <0.001 |
| Model 3 | 156.42 | [49.34–263.50] | 0.004 |

Notes:
Model 1: Linear regression for baPWV and ApoA1.
Model 2: Model 1 and adjusted for age, BMI educational attainment, physical activity, smoking status and history of hypertension.
Model 3: Model 2 and adjusted for HDL LDL.

attainment, physical activity, smoking, and history of hypertension, the overall ApoA1 level in NAFLD patients was positively associated with baPWV (OR, 1.69; 95% CI [1.08–2.66], $P < 0.05$); ApoA1 level in male NAFLD patients was significantly associated with baPWV (OR, 2.80; 95% CI [1.55–5.04], $P < 0.001$). No statistically significant association was observed between female NAFLD patients' ApoA1 level and baPWV. In model 3, adjusted for model 2 and HDL and LDL, the overall ApoA1 level in NAFLD patients was significantly associated with baPWV (OR, 4.46; 95% CI [1.85–10.77], $P < 0.001$); ApoA1 level in male NAFLD patients was significantly associated with baPWV (OR, 3.96; 95% CI [1.29–12.30], $P < 0.05$). No statistically significant association was observed between ApoA1 and baPWV in female patients (Table 2). In the subgroups divided by serum AST/ALT ratio, in model 1, only ApoA1 level in AST/ALT<1 subgroup was significantly associated with baPWV (OR, 1.88; 95% CI [1.06–3.33], $P < 0.05$). In model 3, the overall ApoA1 level in two subgroups was positively associated with baPWV (OR, 4.18; 95% CI [1.16–15.1], $P < 0.05$) in AST/ALT<1 subgroup and (OR, 4.70; 95% CI [1.36–16.23], $P < 0.05$) in AST/ALT ≥ 1 subgroup respectively.

As shown in Table 3, the relationships between ApoA1 and baPWV in male NAFLD patients were analyzed by linear regression. In model 1, the relationship between ApoA1 and baPWV in male patients was found to be significant in univariate analysis (coefficient, 160.02; 95% CI [107.37–212.66], $P < 0.001$). In model 2, after adjusting for age, gender, BMI, education attainment, physical activity, smoking and history of hypertension,

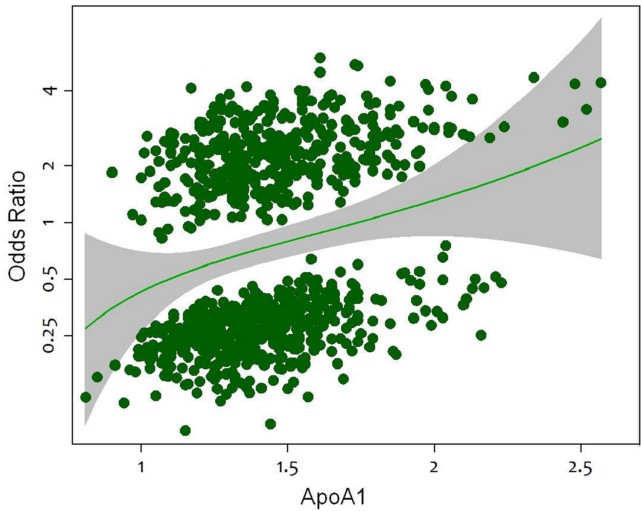

**Figure 1 Relationship between ApoA1 as a continuous variable and risk of the incidence of arterial stiffness in male patients.** Results were obtained by multivariable logistic regression with restricted splines including ApoA1. ApoA1 levels were categorized into 0.5 g/L increment from <1 to >3 g/L and were adjusted for age, gender, BMI, education, physical activity, smoking, history of hypertension, HDL and LDL level. The upper and lower 95% CIs are denoted by a dotted line.

the relationship between ApoA1 and baPWV in male patients was statically significant (coefficient, 139.84; 95% CI [83.97–195.68], $P < 0.001$). In model 3, which adjusted for model 2 and HDL and LDL, the relationship between ApoA1 and baPWV in male NAFLD patients remained significant (coefficient, 156.42; 95% CI [49.34–263.50], $P < 0.05$). ApoA1 is associated with the incidence of arterial stiffness only in male NAFLD patients.

The relationship between ApoA1 and the incidence of arterial stiffness in male patients followed a J-shaped curve, with increased odds ratio at low and high ApoA1. In a multivariable logistic regression model, using ApoA1 as a covariate, adjusting for baseline covariates, the risk for the incidence of arterial stiffness increased with ApoA1 (Fig. 1).

## DISCUSSION

The major finding of this study is that risk of arterial stiffness increased with increasing ApoA1 after adjusting for multi-confounders in NAFLD subjects. The relationship between arterial stiffness and ApoA1 was statistically significant only in male NAFLD population.

The prevalence of NAFLD is likely to rise even further not only in western countries but also in developing countries as China. There is growing evidence that NAFLD must be considered an important independent risk factor for CVD (*Ozturk et al., 2015*). Longitudinal studies have shown that cardiovascular mortality is the most important cause of death in patients with NAFLD (*Byrne et al., 2009*). But the mechanisms by which NAFLD contributes to cardiovascular disease have not yet been fully clarified. In our current study based on a mean age of 48.7 years old population, the mean baPWV is 1,389.48 ± 273.5 cm/s, which is even much higher ($P < 0.001$) than Chinese population with an equal age level ($P = 0.066$) (mean value: 1,340 cm/s) (*Zeng et al., 2016*). NAFLD is

associated with increased baPWV, thus it is of vital importance to identify the risk factors for arterial stiffness in this specific population.

Hepatic steatosis arises from an imbalance between triglyceride (TG) acquisition and removal. Many studies have demonstrated that hepatic TG accretion from increased dietary intake, repartitioning of TG from adipose tissue to the liver, or increased de novo lipogenesis result in hepatic steatosis (*Berlanga et al., 2014*; *Cohen, Horton & Hobbs, 2011*). ApoA1 is the major apolipoprotein in HDL, which plays an important role in fatty acid metabolism (*Yanai et al., 2015*), thus ApoA1 can influence NAFLD by regulating the accumulation of liver triglyceride intake in the diet (*Choe et al., 2013*). ApoA1 plays a central role in the process of reverse cholesterol transport (RCT), which has been thought to be the major mechanism of the antiatherogenic functions of HDL (*Yancey et al., 2003*). In animal models, increasing HDL-C levels did not affect the rates of RCT, and the rate of RCT achieved the maximum at the normal levels of HDL-C (*Alam, Meidell & Spady, 2001*). While infusion of a smaller ApoA1 containing into mice was found to increase reverse cholesterol transport, and over-expression of ApoA1 was found to favor efflux of cholesterol (*Alam, Meidell & Spady, 2001*). At the same time, the liver is the major producers of ApoA1 as well as the important location of reverse cholesterol transport (*Timmins et al., 2005*). Due to the limited studies on the relationship of ApoA1 and NAFLD, further research should be conducted to clarify other potential mechanisms by which ApoA1 could affect the pathologic processes of NAFLD.

According to previous studies, lower ApoA1 may have positive effect on atherosclerosis or high-risk plaque features (*Zhao et al., 2014*), which is exactly in contradiction to our results. As there are few publications focusing on the relationships between ApoA1 and PWV, unanimous conclusions couldn't be drawn before we conducted more long-term large-scale clinical trials.

In our study on NAFLD subjects, men had significantly higher level of baPWV levels and arterial stiffness than women, consistent with the results of some multicenter studies on the general population (*Kim et al., 2014*). Many factors, such as body size, biochemical properties of arteries, sex hormones, can lead to differences in arterial stiffness between male and female (*Janner et al., 2010*; *Rossi et al., 2011*; *Segers et al., 2007*). Some studies showed the distribution of ApoA1 differs between men and women, higher in women than in men (*Csaszar et al., 1991*; *Valles et al., 2002*). A study of 147,576 Swedish males and females aimed at ApoA1 Distribution shown that the ApoA1 concentration was 10% lower than in females (1.36 ± 0.22 g/L vs 1.51 ± 0.24 g/L) (*Jungner et al., 1998*), which is consistent with the National Health and Nutrition Examination Survey III of the United States (*Bachorik et al., 1997*). Therefore, we investigated the relationships between arterial stiffness and ApoA1 in different gender group for the first time. Female NAFLD subjects' serum ApoA1 level is much higher than Male's (1.62 vs 1.42, $P < 0.05$), which is exactly consistent with the results from other populations.The difference in ApoA1 level in different gender groups implied that sex hormones may play an important role in serum ApoA1 level. Both basic and bench research may be designed to test hypothesis and the internal mechanisms.

### Limitation

Fatty liver was diagnosed by ultrasonography rather than liver biopsy pathology. Liver biopsy is the "gold standard" for defining fat accumulation, inflammation and fibrosis in liver, but it is not suitable as a screening method for population-based epidemiological studies due to the invasiveness of liver biopsy. Additionally, this current study only investigated the relationships on NAFLD patients in a cross-section research, to clarify the impact of ApoA1 on CVD risks, further longitudinal study should be performed. Despite these limitations, this study conducted a meaningful study of the relationship between ApoA1 and arterial stiffness in NAFLD subjects.

## CONCLUSIONS

Unlike the protective effect of ApoA1 on atherosclerosis, the serum ApoA1 was associated with arterial stiffness in male NAFLD patients, suggesting that NAFLD may alter arterial stiffness by "ApoA1-related" mechanism in male NAFLD population.

## ACKNOWLEDGEMENTS

We are very thankful the suggestions on our work from Prof. Cruickshank Kennedy and Dr. Luca Faconti's in King's College London.

### Funding

The study was supported by the National Natural Science Foundation of China (81800393), the National Key Research and Development Program of China (2019YFF0216300), the Sub-project of National Key Research and Development Program of China (2018YFC1311302), the Hunan Youth Talent Project (2019RS2014), the Natural Science Foundation of Hunan Province (2018JJ3783), and the Fundamental Research Funds for the Central Universities of Central South University (Grant No. 2018zzts266). The funders had no role in study design, data collection and analysis, decision to publish, or preparation of the manuscript.

### Grant Disclosures

The following grant information was disclosed by the authors:
National Natural Science Foundation of China: 81800393.
National Key Research and Development Program of China: 2019YFF0216300.
National Key Research and Development Program of China: 2018YFC1311302.
Hunan Youth Talent Project: 2019RS2014.
Natural Science Foundation of Hunan Province: 2018JJ3783.
Central Universities of Central South University: 2018zzts266.

### Competing Interests

The authors declare that they have no competing interests.

## Author Contributions

- Xulong Sun conceived and designed the experiments, performed the experiments, analyzed the data, prepared figures and/or tables, authored or reviewed drafts of the paper, and approved the final draft.
- Ruifang Chen analyzed the data, authored or reviewed drafts of the paper, and approved the final draft.
- Guangyu Yan conceived and designed the experiments, performed the experiments, prepared figures and/or tables, and approved the final draft.
- Zhiheng Chen performed the experiments, authored or reviewed drafts of the paper, and approved the final draft.
- Hong Yuan conceived and designed the experiments, performed the experiments, analyzed the data, authored or reviewed drafts of the paper, and approved the final draft.
- Wei Huang conceived and designed the experiments, authored or reviewed drafts of the paper, and approved the final draft.
- Yao Lu conceived and designed the experiments, performed the experiments, analyzed the data, prepared figures and/or tables, authored or reviewed drafts of the paper, and approved the final draft.

## Human Ethics

The following information was supplied relating to ethical approvals (i.e., approving body and any reference numbers):

Ethics approval for the study protocol and data analysis was obtained from the institutional review board of the Institutional Review Board of Third Xiangya Hospital, Central South University (r18030).

## Data Availability

The raw measurements are available as Supplemental Files.

## Supplemental Information

Supplemental information for this article can be found online at http://dx.doi.org/10.7717/peerj.9757#supplemental-information.

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
