# Peer review of "Gender-specific associations between apolipoprotein A1 and arterial stiffness in patients with nonalcoholic fatty liver disease"

_PeerJ, doi:10.7717/peerj.9757_

## Round 0.1 · original submission · Minor Revisions

Although some of the Reviewer's comments are technically listed as minor, the vast majority of their queries and suggestions will significantly improve the clarity of the manuscript and should be incorporated as much as possible into the revision. In particular, Reviewer 2's suggestions on AST/ALT ratio and Apo B/Apo A-I ratio are very relevant and should be included in the revised results if possible as well as the revised discussion.

Reviewer 1 ·

Basic reporting

The paper is overall well-written and organized. Only the style of reference is not always homogeneous

Experimental design

I’ve read with attention the paper of that is potentially of interest. The background and aim of the study have been clearly defined. The methodology applied is adequately described and overall correct

Validity of the findings

The reviewer is not sure about the results obtained and discussed. Infact, it seems not biologically plausible that ApoAI is related to cardiovascular disease in general population, but related to a worse PWV (directly related to CVD risk!) in NAFLD patients. Otherwise, the authors limit their discussion to the fact that it is contrary to what known on ApoAI.

Additional comments

The authors should consider what raised on the validity of the findings

Reviewer 2 ·

Basic reporting

Clear and unambiguous, professional English used throughout.

Literature references, sufficient field background/context provided.

Professional article structure, figures, tables. Raw data shared.

Self-contained with relevant results to hypotheses.


All the points mentioned above were taken into consideration by the authors.

Experimental design

Original primary research within Aims and Scope of the journal.

Research question well defined, relevant & meaningful. It is stated how research fills an identified knowledge gap.

Rigorous investigation performed to a high technical & ethical standard.

Methods described with sufficient detail & information to replicate.


All the points mentioned above were taken into consideration by the authors.

Validity of the findings

Impact and novelty not assessed. Negative/inconclusive results accepted. Meaningful replication encouraged where rationale & benefit to literature is clearly stated.

All underlying data have been provided; they are robust, statistically sound, & controlled.

Conclusions are well stated, linked to original research question & limited to supporting results.

Speculation is welcome, but should be identified as such.

All the points mentioned above were taken into consideration by the authors.

Additional comments

1. Original Submission

1.1. Recommendation

Minor Revision

Manuscript Number: 48102

Title: Gender-specific associations between apolipoprotein A1 and arterial stiffness in patients with nonalcoholic fatty liver disease

2. Comments to Authors:

2.1. Overview and general recommendation:

The aim of this manuscript was to elucidate potential biomarkers for the prevalence of CVD, its related complications and the role of the arterial stiffness in NAFLD patients. Their results showed that serum ApoA1 concentration was associated with arterial stiffness only in male NAFLD patients, unlike in Female NAFLD patients after controlling for multiple risk factors.
The manuscript is well written and original. The scientists have used relevant tools to assess NAFLD and arterial stiffness. Besides, they took into consideration lots of excluding criteria which make the findings more accurate and relevant. The methods section is pretty well described along with the statistical analysis. This work presents interesting and novel results pertaining to the importance of this subject.
This paper has a potential to be accepted, but some important points have to be take into consideration.

2.2. Minor Comments:

1) It is unclear which stage of NAFLD/NASH authors are referring to, is it steatosis along with inflammation or fibrosis is there too. A good assessment would be serum AST/ALT ratio which is usually <1, but can increase as fibrosis advances. Patients can be divided into 2 groups AST/ALT ratio lower than 1 and a group with an AST/ALT ratio higher than 1. As such, they can correlate serum ApoA1 concentration with the specific NAFLD/NASH stage and it can become an important diagnostic tool linking fatty liver to arterial stiffness.

2) The altered (Apo B/Apo A-I) has been suggested to be a good predictor of cardiovascular risk, even better than HDL/LDL ratio. Studies have shown that there is an increase (Apo B/Apo A-I) ratio among patients with NAFLD. Have the authors assessed these parameters to link them with arterial stiffness?

Thus, the paper can be accepted after addressing the above-mentioned comments.

Reviewer 3 ·

Basic reporting

PeerJ paper review:

The is a professionally written tight story that reveals interesting male-female discripances in the association between ApoA1 and arterial stiffness in NAFLD patients in China.

Thanks for referring to the studies that contradict with the present study within the discussion.

The authors use “sex” and “gender” terms within the manuscript. Example: the authors mention in the results section that data were corrected for sex, whereas the main finding related to gender differences. Please note that sex and gender should not be used interchangeably. NIH has distinct definitions for each of these terms. The authors should stick to the term that was tested in the current study.

The current study reports a male-female differenced in baPWV. What does that mean?

The authors bring up an interesting point to the discussion, which the potential male-female difference in ApoA1 distribution. Please expand the discussion to cover how that may provide an explanation for the present findings

In lines 70 and 71: ApoA1 is the major protein component of HDL. Please expand on this central concept for rationalization of the study.

Clarify whether data are presented as mean +/- standard error OR mean +/- standard deviation.

In the results section. Lines 146-199: values are mentioned without inclusion of their standard deviation or error. Please include +/- values within text as within the tables.

In the Discussion. Lines 193-195: how was this statistical comparison performed between the current finding and the previous one? The comparison should not rely only on comparing the mean values without consideration for the variability within the studies.

The English language should be revised to ensure comprehension of the reader (especially for the abtract). For example: there are some missing letters in the abstract page “stiffness, effects”

Abbreviations should be spelled out the first time they are mentioned within the manuscript (Examples include: NAFLD, baPWV, CV, CVD, DBP, SBP, HDL, LDL..)

Many words are written with a capitalized first letter, although this is not needed and they are witten in the middle of a sentence (this is very frequent across the manuscript. examples: Males, Females, Education, …

Experimental design

No comment

Validity of the findings

No comment

---

## Round 0.2 · accepted · Accept

Thank you for the opportunity to review this important work.

Reviewer 1 ·

Basic reporting

The paper is overall well-written and organized.

Experimental design

I’ve read with attention the paper of that is potentially of interest. The background and aim of the study have been clearly defined. The methodology applied is adequately described and overall correct

Validity of the findings

The authors have clarified the validity of their findings.

Additional comments

The authors have considered the reviewers’ suggestion and improved the paper accordingly. I’ve no further comments on it.

Reviewer 2 ·

Basic reporting

all the criteria are met

Experimental design

all the criteria are met

Validity of the findings

all the criteria are met

Additional comments

1. Original Submission: Recommendation: Accepted

Manuscript Number: 48102
Title: Gender-specific associations between apolipoprotein A1 and arterial stiffness in patients with nonalcoholic fatty liver disease

2. Comments to Authors:
The aim of this manuscript was to elucidate potential biomarkers for the prevalence of CVD, its related complications and the role of the arterial stiffness in NAFLD patients. Their results showed that serum ApoA1 concentration was associated with arterial stiffness only in male NAFLD patients, unlike in Female NAFLD patients after controlling for multiple risk factors.
This work presents interesting and novel results pertaining to the importance of this subject.

 The authors did provide a satisfactory answer regarding the AST/ALT ratio comment. Unfortunately, they do not have ApoB values available in their database currently but still the paper can be accepted without that parameter. In addition, they did handle well the crucial comments given by the other reviewers.